# Cow’s Milk Antigens Content in Human Milk: A Scoping Review

**DOI:** 10.3390/foods11121783

**Published:** 2022-06-17

**Authors:** Carlos Franco, Cristina Fente, Cristina Sánchez, Alexandre Lamas, Alberto Cepeda, Rosaura Leis, Patricia Regal

**Affiliations:** 1Department of Pediatrics, Medicine Faculty, Santiago de Compostela University, 15705 Santiago de Compostela, Spain; carlosfrancofente@gmail.com (C.F.); mariarosaura.leis@usc.es (R.L.); 2Department of Analytical Chemistry, Nutrition and Bromatology, Santiago de Compostela University, 27002 Lugo, Spain; crissanchezfente@gmail.com (C.S.); alexandre.lamas@usc.es (A.L.); alberto.cepeda@usc.es (A.C.); patricia.regal@usc.es (P.R.)

**Keywords:** breast milk, human milk, breastfeeding, cow’s milk protein allergy, dietary avoidance, β-lactoglobulin, caseins

## Abstract

The functionality of breast milk in terms of immunity is well-known. Despite this, a significant proportion of breastfed infants exhibit sensitization to different potentially allergenic proteins and clinical reactivity (including anaphylaxis) early in life and before the introduction of complementary feeding for the first time. The potential induction of early oral tolerance to overcome early allergic sensitization through exposure to allergens in breast milk also remains controversial and not yet well-established. The objective of this scoping review is to provide a critical appraisal of knowledge about the content of cow’s milk antigens in human milk. The amount of dietary derived milk antigens found in human milk and the analytical methodologies used to detect and quantify these antigens, the allergic status of the mother, the stage of lactation, the time of sampling (before or after ingestion of food), and the impact of human milk allergen on the infant were the outcomes that were assessed. Allergy risk was explored in all reviewed studies and could help to better elucidate its role in the context of allergic disease development. According to the included literature, we can conclude that there are mainly fragments derived from bovine proteins in human milk, and the presence of potentially allergenic molecules is greater in the milk of mothers with an allergic tendency. A clear relationship between maternal diet and allergen content in breast milk could not be firmly concluded though. Also, infants receiving milk from human milk banks, where donor milk is pasteurized for preservation, may be subject to greater risk of allergy development, especially for β-lactoglobulin.

## 1. Introduction

Breastfeeding is the natural way of feeding infants during their first six months of life, and its continuation is thereafter recommended by health agencies, together with complementary feeding, up to 1 year of age or beyond [1]. Human milk is a nutritionally well-balanced food, and in addition, it contains a wide variety of functional components that modulate the growth and development of the baby [2]. Its functionality is also evident in terms of immunity, delivering a wide range of immunologic factors to the immature immune system of the newborn, including several of the components of the fat globule oligosaccharides; a very diverse collection of proteins, including immunoglobulins; cytokines, growth factors, and microorganisms [3]. There are many studies that show the benefits of breastfeeding for the infant include a reduced risk of developing atopy, asthma, and other immune-mediated diseases. Breastfeeding protects against the development of atopic disease, and this effect appears even stronger in children with atopic heredity [4,5]. The American Academy of Pediatrics Committee on Nutrition published a clinical update in 2019 that provided information on the relationship between breastfeeding and atopy, concluding that breastfeeding provides a protective effect against atopic dermatitis by at least during the first two years of life [6]. In this sense, a randomized clinical trial evaluating the short-term efficacy of topical application of human breast milk versus hydrocortisone 1% ointment in infants with mild to moderate atopic dermatitis asserts that human milk provides the same results in the healing of this dermatitis [7].

Aside from the above, formula-fed infants receive high amounts of antigens from bovine milk, which is the basis for most infant formula, while for breastfed babies, the first exposure to food allergens is breast milk, in which the antigens from the maternal diet and presumably also those inhaled by the mother will be found in much smaller amounts than in infant formula [8]. The presence of food antigens in breast milk will depend on factors such as maternal digestion and transfer rate to the mammary gland. A significant proportion of infants already exhibit sensitization to different potentially allergenic proteins and clinical reactivity (including anaphylaxis) early in life and before introduction of complementary feeding [9]. However, it has not been well-documented yet whether they can trigger allergic sensitization in infants. The latest systematic reviews on this topic [10,11,12,13,14] as well as the 2019 American Academy of Pediatrics report [6] concluded that there is a lack of evidence to support maternal dietary restrictions during pregnancy and lactation. After many years of allergen avoidance, new guidelines recommend early exposure of the infants to oral allergens with the goal of inducing regulatory immune responses and generating tolerance to allergens [6]. However, the potential induction of early oral tolerance to overcome early allergic sensitization through exposure to allergens in breast milk also remains controversial and not yet well-established [15]. According to the latest studies using mass spectrometry, the non-human proteins most frequently found in breast milk are those that come from bovine milk [16,17]. The goal of this study was to conduct a scoping review to assess the existing knowledge on the bovine antigens content of breast milk. The study of the variations between mothers, lactation stage, quantification of antigens before and after maternal consumption, and analytical methodology could help to better elucidate their role in the context of the development of allergic diseases. The clinical relevance of these proteins in the induction of allergic sensitizations in breastfed babies would be overviewed as well. 

## 2. Methodologies

### 2.1. Search Strategy and Inclusion/Exclusion Criteria

To synthesize the evidence and assess the scope of the literature on the topic, a scoping review was conducted in accordance to the PRISMA Extension for Scoping Reviews (PRISMA-ScR) approach [18]. The following six methodological steps are described in more detail below: identifying the research question; identifying relevant studies; selecting relevant studies; graphing the data; collecting, summarizing, and reporting the results [19].

The main research question was “What is known about the content of cow’s milk allergens in human milk and the factors that may influence their presence?”

A comprehensive literature search of *PubMed*, *Scopus*, and *Web of Science Core Collection* was performed in January 2022, and it was limited to articles published in English since 1980. Text words and controlled vocabulary for four concepts were used and searched within the titles, abstracts, and keywords of articles: human milk, cow’s milk dietary antigens, dietary avoidance, and food allergy. Specific immunoglobulins were not part of this review.

Original research papers were revised to investigate the transfer of bovine milk dietary allergens to infants through breast milk and the factors that may influence their presence. The outcomes that were assessed included: amount of diet-derived dairy antigens found in human milk and analytical methodologies used to detect and quantify these antigens, mother allergic status, lactation stage, sampling time (before or after food ingestion), and impact of allergen shedding in human milk on baby allergy risk. Studies published as full-length articles were selected, excluding conference abstracts, books, editorials, and letters to the editor. Randomized controlled trials, prospective, cohort studies, intervention studies, and cross-sectional observational studies examining breastfed infants were screened. Reviews were excluded.

### 2.2. Article Screening and Data Abstraction

Titles and abstracts of all papers were assessed for their potential relevance according to the inclusion and exclusion criteria explained above. The screening was performed independently by C.F. (Carlos Franco) and C.F. (Cristina Fente) Any discrepancies were resolved by consensus. Data were extracted from the full text of the selected papers and subsequently reviewed by C.F.F., C.F.S., C.S., P.R., and A.L. Studies were initially appraised individually before comparing and summarizing the findings, searching for links between breast milk production circumstances, breast milk allergen content, and infant allergic sensitizations.

## 3. Results

### 3.1. Synthesis

A total of 230 records were identified in the databases applying the exclusion and inclusion criteria and after duplicate removal. Once the title and abstract had been studied, 50 papers were submitted to the full-text evaluation. In total, 27 articles were included in this scoping review. Final search results are shown in the PRISMA flow chart (Figure 1).

For all studies gathered in this review, data abstraction details are summarized in Table 1, including information on maternal diet, population details (allergic mother or not) and number of samples, cow’s milk antigen, lactation stage, sampling time after cow’s milk intake, analytical methodology, method sensitivity, antigen maximum level, number of samples with detectable levels of antigen, and clinical relevance.

### 3.2. Dietary Cow’s Milk Allergens Found in Human Milk

Cow’s milk contains 30 to 35 g/L of protein, of which approximately 80% are caseins, and 20% whey proteins [44]. β-lactoglobulin (β-LG) is a whey protein strongly associated with cow’s milk allergy, but caseins and all other milk proteins could potentially cause allergy as well [44]. As pointed out by this review, the two dietary cow’s milk allergens that are monitored in targeted studies of human milk are β-LG and αS1-casein, the major milk allergens [45]. β-LG is a globular protein that naturally occurs as a dimer of a 18 kDa monomer unit and that resists well the acid and enzymatic digestion in the human digestive tract [44]. Caseins are more abundant than whey proteins in bovine milk but are degraded very early in the stomach and then in the intestine [46]. 

Most of the reviewed studies on human milk searched only bovine β-LG [20,21,22,23,24,25,27,28,29,30,31,32,33,34,35,36,42]. Other researchers investigated both β-LG and αS1-casein [37,38,41], while Pastor-Vargas [39] monitored β-LG along with other food allergens. Untargeted studies were also included in this revision, showing that derived peptides from bovine proteins, notably from bovine caseins and β-LG, are also found in human milk [16,17,40,43].

### 3.3. Analytical Methods and Maximum Detected Levels of Cow’s Milk Antigens in Human Milk

A recent review has summarized the many instrumental techniques applied in several areas of the food allergy field [47]. Nonetheless, specific surface plasmon resonance (SPR) sensors have been just developed to detect casein and β-LG [48], and its potential application to human milk is undeniable. In the present review, only the analytical techniques used for the detection of potentially allergenic proteins present in breast milk with unequivocal origin of cow’s milk have been included. Analytical possibilities to determine cow milk antigens (intact proteins and/or peptides) include immunochemical methods. Four studies using radioimmunoassay techniques [21,22,24,25] detected β-LG in human milk samples in concentrations between 4.4 and 800 ng/mL, the latter being the maximum reported level so far. ELISA techniques have been employed by several authors [17,20,23,26,27,28,29,30,31,32,33,34,35,41,42]. The maximum value detected for β-LG in human milk using this technique was 150 ng/mL [27]. β-LG detection but not quantification was performed also using antibody microarray technology [39]. When the results obtained with other types of analytical approaches are compared, the main concern from cross-reactivity or the recognition of non-specific antibodies becomes evident. Bertino et al. [32] found β-LG in human milk at a maximum level of 87.5 ng/mL by ELISA determination, but the results were not corroborated using protein purification methodologies followed by microsequencing. Furthermore, Picariello et al. [41] found β-LG and αS1-casein using ELISA, but the intact protein was not detected by SDS-page and parallel Western blot analysis. Intact bovine αS1-casein was found in human colostrum using ProteoMiner technology followed by 2D-PAGE and nano-LC-MS proteomic techniques [37,38]. In the last decade, LC-MS-based proteomics has been applied by several authors [16,17,40,41,43] to detect bovine milk proteins in breast milk, subjecting samples to enzymatic digestion and measuring the resulting peptide mixtures. The inconsistency of the ELISA results was demonstrated by Picariello et al. [41] with only one positive sample collected from a from a woman on a strict milk- and dairy-products-free diet. The results using Western blotting with anti-β-LG antibody confirmed the absence of intact β-LG while with LC-MS/MS detected a αS1-casein fragment and two peptides from β-LG in the milk of lactating mothers who drank bovine milk. The same authors [17], in order to target possible intact cow’s milk proteins, analyzed the resulting tryptic digests from the 12% TCA-insoluble protein pellets that were Cys-reduced/alkylated in denaturing buffer (6M guanidine) by HPLC-MS/MS. The search of a database containing cow’s milk protein sequences using both trypsin cleavage specificity or non-specific cleavage suggests shifting the analytical perspective for the detection of dietary food allergens in breast milk from intact proteins to digested peptide fragments. SDS-PAGE and parallel Western blot analysis confirmed the substantial absence of intact β-Lg in the analyzed samples. LC-MS/MS runs (targeted ion extraction) specifically aimed at detecting the sequences identified by Zhu et al. [43] were not able to detect any of the bovine αs1-casein peptides generated by tryptic digestion of the protein pellet. Inter-individual variability of the excretion can explain this finding. Proteins that are likely from bovine milk products (α-S1-, α-S2-, β-, κ-caseins, and β-LG) were the dominant nonhuman proteins in the Dekker et al. study [16]. Bovine α-S1-casein was found with nearly identical peptide sequences. This remarkable reproducibility and distribution of the identified peptides provides credence that this protein likely is present intact in human milk.

### 3.4. Cow’s Milk Allergens Levels Found in Human Milk and Maternal Diet 

Many of the studies included in this review were aimed at verifying whether there are differences in the allergen content in human milk between mothers who follow diets free of cow’s milk and mothers who consume this food or its derivatives [17,21,22,23,25,27,30,31,32,41,42]. Basal samples were collected after different periods of cow’s milk avoidance, ranging from 24 h to 4 weeks depending on the study (see Table 1). Detectable levels of β-LG were found in the basal samples by several authors [22,23,25,26,30,32,42] up to 7 or 10 days after maternal consumption of cow’s milk [32,42] or even 4 weeks later [26].

The amount of food and the period that elapses between its intake and sample collection is also variable among the reviewed studies. The influence of cow’s milk serving size (ranging between 200 and 1500 mL) on the quantities of allergen detected in breast milk could not be demonstrated in two studies [24,32]. However, the results obtained by Fukushima et al. [33] suggested that β-LG concentrations in breast milk are related to long-term consumption of cow’s milk. These authors found that the consumption of whey hydrolysate formula by lactating women over a considerable time reduces the transfer of β-LG into their milk and that this low level can be maintained even after inadvertent ingestion of cow’s milk [33,34]. Likewise, Monti et al. [25] inferred an increase in β-LG with a diet rich in cow’s milk proteins. Regarding the moment when the amount of allergen is greater after maternal ingestion of cow’s milk, between 4 and 6 h after ingestion, the maximum level was detected [21,26,27,35]. No correlation was found between the type of milk preparation (homogenized or unhomogenized) and breast milk allergen content [26,27]. As for the heat treatment of milk or dairy products, only in Fukushima’s work [34] it is mentioned that it is raw milk. Therefore, we cannot establish a comparison regarding the influence of the thermal treatment on the transfer of the antigen to breast milk.

Other papers included in the present review studied mothers following a non-restrictive diet, i.e., ingesting unknown amounts of milk and dairy products. The results obtained indicate that there is high interindividual variation in dietary protein secretion [16,20,28,36,37,38,39,40,43].

### 3.5. Dietary Allergens Levels Found in Human Milk and Lactation Stage

The presence of allergens in colostrum [23,24,28,37,38] and transition milk [21,24] have been evaluated in some of the studies included in this review. No higher allergen levels were found in colostrum compared to transition milk [24] nor when transition milk was compared to mature milk [21]. In two papers [37,38], colostrum from mothers with premature or full-term delivery were evaluated, with findings of a higher concentration of αS1-casein in preterm mothers. 

### 3.6. Dietary Allergens Levels Found in Human Milk and Mother or Child Allergic Condition

Several studies have evaluated the level of allergens found in milk in connection to the allergic condition of the mother [16,24,26,27,28,30,31,40,42]. No correlation was found between maternal allergic condition and breast milk concentration of β-LG by some authors [24,27,28,42]. This was not so for others, in whose studies β-LG levels were higher in atopic mothers [30,31,40].

Regarding untargeted approaches, some proteins differed significantly in concentration between the breast milk of allergic and non-allergic mothers. Protease inhibitors and apolipoproteins were present in much higher concentrations in breast milk of allergic than non-allergic mothers. A link between these proteins and allergy and asthma has been suggested before [40]. A significant difference in levels of non-human proteinaceous molecules in human milk of allergic and non-allergic mothers has been observed, and this difference can be largely attributed to sequences that match to bovine proteins β-LG and α–2-HS-glycoprotein [16].

Two studies compared β-LG levels in human milk from mothers with allergic offspring with those from mothers of healthy infants [26,28], finding no relationship between the child’s allergic condition and the cow’s milk allergen content in breast milk. Other authors [28] concluded that the milk of the mothers whose infants became allergic to cow’s milk contained less IgA through the lactation, and the difference was most marked in the colostrum. Nonetheless, β-LG measurements were similar.

### 3.7. Clinical Relevance of Dietary Allergens Levels Found in Human Milk

Many of the reviewed studies show that the presence of allergens in breast milk is related to obvious clinical manifestations in children. Thus, milk from mothers whose infants suffered from infantile colic contained high amounts of β-LG, and the offspring of mothers with non-detectable amounts became free from colic [22]. Allergy incidence in the infants born in the atopic diet group with milk-free diet was significantly lower compared with that of the atopic group on the unrestricted diet [31]. All basal samples containing detectable amounts of β-LG were collected from mothers of infants with cow’s milk protein allergy (CMPA), and no β-LG was found in basal samples from mothers with infants without CMPA [30]. Most of the infants with CMPA reacted to cow’s milk challenge through human milk, and β-LG concentration between 0.01–11.54 ng/mL could exacerbate symptoms [35]. The presence of symptoms in the infant, such as diarrhea, vomiting, colic, or exanthema, was significantly correlated to high levels of β-LG in breast milk [24]. Breastfeeding with milk containing β-LG elicited symptoms of CMPA in three of the four CMPA infants [42].

Only one study could not find a clear relationship between the presence or amount of β-LG in breast milk and the allergic symptoms in the offspring. The presence or quantity of β-LG was unrelated to breast-milk antibody levels, infantile symptom scores, or maternal atopic history [23]. 

## 4. Discussion

Breast milk can promote tolerance induction in off-spring in two ways: instructing long-term immunity through antigen presence [49] or promoting immune regulation mechanisms in a non-antigen specific way [50]. However, the variables that control heterogeneity in oral tolerance induction and allergy prevention in children breast-fed are not clear [49]. Breast milk is a complex matrix that contains an impressive array of bioactive components that guide the development of the immune system and leave their mark on improved health even in adulthood [51]. In addition to endogenous proteins, which are very important from a nutritional and immunological point of view [52], the presence in human milk of intact proteins or biologically active peptides derived from the mother’s diet has been studied for decades. 

Factors affecting antigen digestion and absorption across the gut barrier are key [49]. Upon ingestion, most protein breaks down into its amino acids or small peptides in the digestive system, reducing its allergenicity. However, for a variety of reasons, including anti-acid treatments [53] or individual variability, a small proportion of intact proteins or their peptide fragments may resist digestion, and these intact proteins could pass the intestinal barrier and appear intact in the bloodstream. The existence of the intestine-breast axis can explain the presence of these non-human molecules in human milk, but the permeability of the mammary gland is also decisive. The transfer of proteins through the intestinal epithelium protected by forming immunocomplexes of IgG antigen through the neonatal Fc receptor (FcRn) [54], which is also expressed in the epithelium of the human mammary gland [55], can facilitate the transfer of circulating antigen-immune complexes from the serum to the mammary gland. In some circumstances, such as preterm birth, the first stages of lactation, or discontinuous lactation, the permeability of the mammary epithelium increases and, accordingly, also the possibilities that these molecules appear in breast milk [56]. Intestinal permeability can be increased in subjects with pathologies such as celiac disease or atopy [45]. Nevertheless, regarding the enhancement of the immune system or, on the contrary, the increase in allergic susceptibility, which would result from the presence of these dietary proteins or their fragments in breast milk, is still a controversial issue in the scientific community [49].

We will begin this discussion at this last point. Regarding the clinical relevance for breastfed infants of the presence of cow’s milk allergens in the breast milk of their mothers, six studies in this review reported that maternal ingestion of β-LG triggered reactions in infants ranging from colic [22,57] and other digestive symptoms such as diarrhea or vomiting, rash [24], or an exacerbation of symptoms in infants with CMPA [42,57]. In one of the studies, all baseline samples containing detectable amounts of β-LG were collected from mothers of infants with allergy to cow protein [30], and in another, the incidence of allergy in infants born from atopic mothers on a diet without restrictions was significantly higher [31]. Only Machtinger and Moss [23] couldn’t find a clear relationship between the presence or amount of β-LG in breast milk and the allergic symptoms in the offspring. In view of these results, we could infer a direct relationship between the intake of food allergens by the mother and the triggering of symptoms in the baby can be inferred. However, to reach this conclusion, it is necessary to be sure that it is a causal and not a casual association. In a review of the existing guidelines for allergy to cow proteins published between 2012 and 2019 [58], it was pointed out that although many of the guidelines recommend the strict exclusion of cow’s milk in maternal diet to control symptoms in breastfed babies, clinical trials do not provide consistent support for this recommendation. The reactions observed could be due to normal behavior in an infant with possible food allergies and not necessarily due to the presence of allergens in breast milk.

Subsequently, we must then ask ourselves if the food allergens that lactating women ingest are found in breast milk at levels that exceed the thresholds of reactivity to induce an allergic response. Most of the papers included in this review searched for allergens before consuming cow’s milk. The presence of β-LG was observed in milk samples 7 days after maternal consumption of cow’s milk [42] or 10 days later. Conversely, Bertino et al. [32] found no differences in the amount of allergen in mothers who consume cow’s milk or not, concluding that this observation was due to interference in the ELISA method. The influence of the amount of cow’s milk consumed was not demonstrated in some studies reviewed [24,32]. Still, other authors suggested that the level of bovine antigens is related to the long-term consumption of cow’s milk [33]. In mothers with a non-restrictive diet, the results obtained indicate that there are high inter- and intra-individual variation in dietary protein secretion [16,17,36,37,38,39,43]. Cow’s milk and dairy products are subjected to a series of processing methods that may affect not only their immunogenic activity but also the amount of antigens that are effectively absorbed in the intestine of the lactating mother [59]. Milk proteins are unstable when heated, leading to modifications in structure and allergenicity, and microbial fermentation with lactic acid bacteria induces proteolysis, increasing the tolerability of cheese and yogurt in clinically reactive patients. Conversely, a recent study conducted using a murine model demonstrated that raw (unprocessed) cow’s milk and native whey proteins have a lower allergenicity than their processed counterparts [60]. The human proof-of-concept provocation pilot conducted in parallel to the animal study also provided evidence that milk processing negatively influences the allergenicity of milk, possibly due to the release of peptides with allergenic potential. In particular, heat treatments lead to the formation of protein aggregates, which may be an important factor to consider regarding uptake of milk proteins by Peyer’s patches and sensitization. In an elaborate study by Roth-Walter et al. [61], native β-lactoglobulin was readily transcytosed through enterocytes in vitro and in vivo (in mice) but not casein. Casein forms micelles and does not efficiently cross the epithelial barrier. Conversely, pasteurized milk leads to aggregation of whey proteins but not of caseins, deviating the absorption of the first to Peyer’s patches and hence to the sensitization route. The oral intake of soluble proteins/fragments would induce anaphylaxis but not aggregates [61]. Breast milk is consumed raw by the breastfed infant, but the situation changes dramatically when the infant is receiving milk from human milk banks, where pasteurized donor milk is provided to infants in need. Considering the importance of aggregates in sensitization, the presence of cow’s milk antigens in human milk would be responsible for allergy development in infants fed with pasteurized donor milk but not for exclusively breastfed infants, especially for b-LG. In the case of children already sensitized by processed food, the content of bovine proteins in breast milk could induce anaphylaxis since it is consumed raw. Apart from temperature, the fat content of food has also shown effects on allergen bioavailability by reducing it [62]. Similarly, previous results obtained with peanuts indicate that thermal processing applied to food has an influence on its allergological properties due to extensive protein fragmentation [63]. In view of these results, a clear relationship between maternal diet and allergen content in breast milk cannot be founded.

The maturation of the mammary gland determines the stages of lactation and the existing differences in the composition of breast milk in preterm births. In the first 72 h after delivery, colostrum is produced, a fluid in which we find components that enter the interior of the luminal space of the mammary gland through the transcellular route from the maternal blood capillaries. During the following 15 days, the mammary gland reaches maturity, the intercellular junctions in the mammary epithelium close, and milk secretion is produced by transcellular synthesis and transport [64]. These changes could also determine differences in the presence of allergens. The greater passage of allergens into breast milk in colostrum from preterm versus term colostrum samples can explain the differences in αS1-casein content found in two studies [37,38]. However, similar levels of allergens were found at different stages of lactation by other authors [21,24]. 

The relationship between the allergic status of the mother, which could lead to increased intestinal permeability in these women, and the presence of allergens in her milk was also explored. Most of the reviewed studies found significant differences [16,24,26,27,30,31,40,42] also in atopic women [30,31,40]. Even untargeted studies based on HPLC/MS-MS methods showed that milk from allergic and non-allergic women differed significantly in the concentration of non-human proteins or peptides [16,40]. The exposure to a major cow’s milk allergen through breast milk of mothers with different immune status could influence food allergy outcome in offspring.

As to whether these dietary allergens ingested by lactating women are indeed found in breast milk at levels that exceed thresholds of reactivity to induce an allergic response, only a few samples from three studies [24,25,27] reached a level of β-LG ≥ED01 (eliciting dose for 1% of allergic individuals), and in no case did the samples contain β-LG in concentrations ≥ED05 [65]. The probability of having enough β-LG in breast milk to trigger an allergic reaction in a susceptible child has been estimated at 1:2893 [8]. However, in the previous systematic review, only immunochemical methods were used (ELISA and radioimmunoassays). Bertino et al. [32] failed to reproduce the results obtained with ELISA when analyzing β-LG in breast milk using protein-purification methodologies followed microsequencing. Authors suggested that the quantitative evaluation of bovine β-LG in human milk by ELISA could give rise to misleading interpretations. Pastor-Vargas et al. did not quantify this intact protein using antibody microarray technology either [39]. The inconsistency of the results obtained with immunochemical methods was also demonstrated when samples previously positives from β-LG in ELISA were not confirmed by HPLC-MS/MS [41]. Immunochemical detection requires specific antibodies for the detection of specific proteins. HPLC-MS/MS techniques have been used for non-targeted analysis of cow’s milk proteins or their derived fragments in human milk samples [16,17,41,43]. In all these works, the presence of non-human peptide molecules (but not intact proteins) was confirmed but at very low levels (ppb levels or lower). Even to be able to analyze them, the previous enrichment of the sample is mandatory [17]. The latest investigations suggest a new analytical perspective for the detection of food allergens in human milk from intact proteins to digested peptide fragments.

## 5. Conclusions

The presence of cow’s milk allergens in human milk has been well-documented for decades. However, the amounts detected do not appear to be sufficient to trigger an allergic reaction in susceptible infants. Furthermore, when analytical data are first obtained by immunochemical methods and then contrasted with those obtained by confirmatory MS/MS methods, the results are not reproducible. In this sense, the most recent studies based on proteomics and mass spectrometry approaches suggest that protein-derived peptides rather than intact proteins can be found in human milk. Using modern techniques, it has been documented that the presence of potentially allergenic molecules is greater in the milk of mothers with an allergic tendency. However, we cannot be sure that this fact is related to allergic sensitization or tolerance in infants. The main knowledge gaps around this topic are the real sensitization capacity of allergens and their fragments present in breast milk, the influence of environment, and the cross-reactivity of breast milk and complementary feeding. Further studies are needed linking bovine milk dietary allergens in breast milk with maternal allergic status, stage of lactation, sampling time (before or after food intake), and the impact of elimination of allergens in human milk on infant allergy risk using truly confirmatory analytical techniques.

## Figures and Tables

**Figure 1 foods-11-01783-f001:**
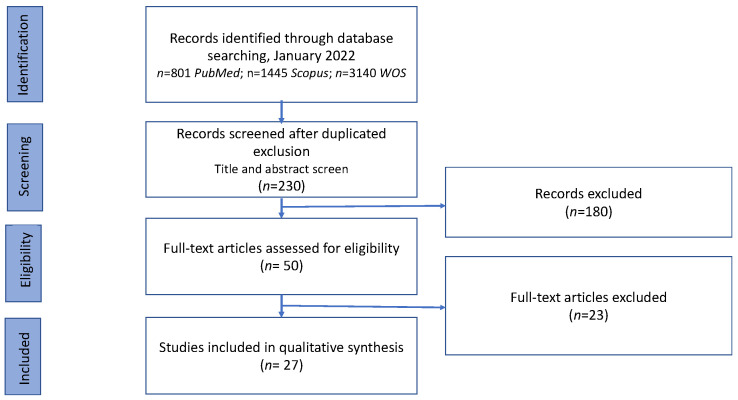
Flow diagram describing study selection process.

**Table 1 foods-11-01783-t001:** Characteristics and findings of studies evaluating the presence of cow’s milk allergens in breast milk (*n* = 27).

Ref.	Maternal Diet	Population (Mothers), Allergic Condition, and Lactation Stage	Sampling Time After Cow’s Milk Ingestion	Antigen, Analysis Method, and Sensitivity	Maximum Level, Sampling Time, Samples with Antigen	Main Finding
Stuart, 1984 [20]	Unrestricted diet	*n* = 28, transition milk	N/R	β-LG, ELISA, N/R	19 ng/mL, 5/28	ELISA may be a simple and useful method for β-LG analysis.
Kilshaw, 1984 [21]	Cow’s-milk-free diet (24 h) or not (500 mL)	*n* = 19, transition and mature milk	2 h, 4 h, 6 h	β-LG, RIA, 0.1 ng/mL	6.4 ng/mL 4 h, 10/19	β-LG was detected in breast milk, but immune complexes were not present in breast milk.
Jakobsson, 1985 [22]	Cow’s-milk-free diet or not	*n* = 38, mature milk	0, N/R	β-LG, RIA, 5 ng/mL	33 ng/mL, 18/38	Milk from mothers whose infants suffered from infantile colic contained high amounts of β-LG. Mothers with non-detectable amounts have infants became free from colic.
Machtinger, 1986 [23]	Cow’s-milk-free diet (*n* = 1) or not (*n* = 56)	*n* = 57, colostrum and mature milk	1, 2, 3, 4, 5, and 6 days after cow’s milk cessation.	β-LG, ELISA	>6.4 ng/mL, N/R, 24/54	β-LG persisted up to 3 days after maternal dietary milk exclusion. The presence or quantity of β-LG was unrelated to breast milk antibody levels, infantile symptom scores, or maternal atopic history.
Axelsson, 1986 [24]	Diet with cow’s milk (200–1500 mL/d)	*n* = 6 (with allergic symptoms), *n* = 19 (without allergy)	N/R	β-LG, RIA, 5 ng/mL	800 ng/mL, 19/25	No correlation was found between mother allergic condition or daily cow’s milk intake and concentration of β-LG. Symptoms in the infant such as diarrhea, vomiting, colic, and exanthema were significantly correlated to high levels of β-LG. The two mothers with detectable β-LG in all milk samples had the highest serum values, and their infants suffered from gastro-intestinal symptoms, weight decline, and exanthema.
Monti, 1989 [25]	Cow’s-milk-free diet (3 days) or not	*n* = 4, mature milk	4 h	β-LG, RIA, N/R	415 ng/mL, 4/4	Immunoreactivity was positive even in milk from mothers consuming a diet free of cow’s milk. An increase with a diet rich in cow’s milk proteins was detected. The human milk fraction cross-reacting with anti-bovine β-LG antibodies corresponds to the 20 kDa fragment from the N-terminal end of human lactoferrin.
HØST, 1988 [26]	Cow’s-milk-free diet (4 weeks) or not (500 mL)	*n* = 9, with allergic children and *n* = 10, with healthy infants, mature milk	0, 4 h	β-LG, ELISA, 0.3 ng/mL	10.5 ng/mL (0) and 45 ng/mL (4 h), 3/9	No correlation was found between β-LG content in breast milk and cow’s milk ingestion.
HØST, 1990 [27]	Cow’s-milk-free diet (7 d) or not (500 mL homogenized or not cow’s milk alternative each week-)	*n* = 10, healthy and *n* = 10 atopic, mature milk	0, 4, 8, 12, 24 h	β-LG. ELISA, 0.3 ng/mL	150 ng/mL, 4 h, 9/10 (atopic mothers), 10/10 (non-atopic mothers)	No correlation was found between the type of milk preparation (homogenized or not) and the presence of β-LG or the level of β-LG in human milk. The presence of B-LG in human milk is a common finding in both atopic and non-atopic mothers.
Savilahti, 1991 [28]	Unrestricted diet	*n* = 44, colostrum, and mature milk	N/R	β-LG. ELISA, 0.1 ng/mL	33 ng/mL, 26/44	An infant is more likely to develop cow’s milk allergy if the mother’s colostrum had a low total IgA content. β-LG measurements were similar in colostrum and mature milk.
Mäkinen-Kiljunen, 1992 [29]	Cow’s-milk-free diet or not (400 mL)	*n* = 3, mature milk	0, 1, 2, 4, 6, 8, 20 h	β-LG, ELISA, 0.002 ng/mL	4.4 ng/mL, 2 h, 3/3	Trace quantities of bovine β-LG in human milk can be assayed dependably with the method.
Sorva, 1994 [30]	Cow’s-milk-free diet (24 h) or not (400 mL, fat-free)	*n* = 28 non atopic, *n* = 25 atopic, mature milk	0, 1, 2 h	β-LG, ELISA, 0.002 ng/mL	3.5 ng/mL, 0, 23/47; 7.84 ng/mL, 1 h, 39/52	All basal samples with β-LG were from mothers of infants with cow’smilk allergy. Not detected in the basal samples from the mothers with infants without cow’s milk allergy. β-LG was found in the 1 or 2 h samples in 75% of the mothers. β-LG levels were increased in the 1 or 2 h samples as compared with the basal levels in about half of the mothers.
Lovegrove, 1996 [31]	Cow’s-milk-free diet (*n* = 10, atopic) or not (*n* = 24, atopic or not) >500 mL	*n* = 22, atopic, *n* = 12, non-atopic, mature milk	N/R	β-LG, ELISA, 0.08 ng/mL	5.9 ng/mL, 24/24	Women with milk-free diet have significantly lower levels of β-LG than the atopic group on the unrestricted diet. The allergy incidence in the infants born in the atopic diet group was significantly lower compared with that of the atopic group on the unrestricted diet.
Bertino, 1996 [32]	Cow’s-milk-free diet (10 d) or not (200 or >500 mL)	*n* = 14, healthy non-atopic, mature milk	0, 12 H	β-LG, ELISA, SDS-Page, WB, immunostaining, RP-HPLC, sequencing, 0.1 ng/mL	Cow’s-milk-free diet: 86.1 ng/mL, 200 mL: 87.5 ng/mL, >500 mL: 18.5 ng/mL, 14/14	At least in healthy subjects, false-positive results in ELISA determinations of bovine β-LG in human milk might be due to cross-reactions between polyclonal antibodies and different protein antigens.
Fukushima, 1997 [33]	Diet with whey hydrolysate formula, 200 mL (MOM group), or cow’s milk, 200 mL (COW group) (>4 months). Diets switched for the second sampling.	*n* = 12 (MOM group), *n* = 13 (COW group), mature milk	1. 3, 4, 8, 9, 15 h	β-LG, ELISA, 0.1 ng/mL	16.5 ng/mL, first sampling: 2/12 (MOM group) and 11/13 (COW group), second sampling: 3/12 (MOM group) and 8/13 (COW group)	Long=term consummation of cow’s milk increases β-LG in the breast milk. Hydrolysate peptides can be detected in β-LG ELISA. The consumption of whey hydrolysate formula over a considerable time reduces the transfer of β-LG into their breast milk, and the low level can be maintained even after inadvertent ingestion of cow’s milk.
Fukushima, 1997 [34]	Unrestricted diet (200 mL/d, without heating the milk, for 7 d before the sampling day) and then 200 mL cow’s milk the morning of the sampling day.	*n* = 24, healthy, mature milk	1–3 h, 4–8 h, and 9–15 h	β-LG, ELISA, 0.1 ng/mL	16.5 ng/mL, 4–8 h, 15/24	Transfer of β-LG into breast milk was influenced by the maternal consumption of cow’s milk. This result suggests that β-LG concentrations in breast milk are related to long-term consumption of cow’s milk.
Järvinen, 1999 [35]	Cow’s-milk-free diet (2–4 weeks) or not	*n* = 16 (infant with CMA), *n* = 10 (healthy infant), mature milk	0, 1, 2, 3, 4 h	β-LG, ELISA, 0.002 ng/mL	11, 54, 6 h, 13/26	Most of the infants with CMA reacted to cow’s milk challenge through human milk. β-LG could exacerbate symptoms.
Restani, 2000 [36]	Unrestricted diet.	N/R	N/R	β-LG, SDS-PAGE, N/R	N/R	The presence of β-LG in breast milk was not confirmed. The conflicting results reported in the literature about the presence of this bovine protein in human milk are due to cross-reactivity with human proteins. Components other than bovine β-LG or caseins could be involved in the induction of allergic symptoms in exclusively breast-fed children.
Coscia, 2012 [37]	Unrestricted diet including cow’s milk and derivatives	*n* = 62 (term infant), 11 (preterm infant), healthy, colostrum	Just after cow’s milk consumption	αS1-casein, β-LG, Proteomic Techniques, N/R	N/R	Bovine a-S1 casein is secreted in human milk at higher concentration in preterm mothers. A possibility could be the different membrane permeability observed in mothers who delivered prematurely.
Orru, 2013 [38]	Unrestricted diet including cow’s milk and derivatives	*n* = 62 (term infant), *n* = 11 (preterm infant), healthy, colostrum	Just after cow’s milk consumption	αS1-casein, Proteomic Techniques, N/R	N/R	Higher concentration of bovine a-S1 casein in preterm colostrum.
Pastor-Vargas, 2015 [39]	Unrestricted diet	*n* = 14, healthy, mature milk	N/R	β-LG and other 9 major allergens and 4 panallergens, antibody microarray technology, 1 ng in 35 µL	N/R, 13/14	Milk allergens are low; their presence in breast milk is due probably to food ingestion from the mother diet.
Hettinga, 2015 [40]	Unrestricted diet	*n* = 10, allergic, *n* = 10, non-allergic, mature milk	N/R	Non-targeted Proteomic Analysis, LC/MSMS, N/R	N/R	Nineteen proteins, from total of 364 proteins identified in both groups, differed significantly in concentration between the breast milk of allergic and non-allergic mothers. Protease inhibitors and apolipoproteins were present in much higher concentrations in breast milk of allergic than non-allergic mothers. These proteins have been suggested to be linked to allergy and asthma.
Picariello, 2016 [41]	Cow’s-milk-free diet (6 d) or not (200 mL)	*n* = 12, non-atopic healthy, mature milk	0, 2 h	β-LG and αS1-casein intact and hydrolyzed, ELISA, SDS-Page and parallel WB and HPLC/HRMS, 0.1 ng/mL	N/R, β-LG: 2/6 (ELISA) and 0/6 (HPLC/HRMS); N/R, αS1-casein 1/6 (ELISA) and 0/6 (HPLC/HRMS);	αS1-casein fragment and 2 peptides from β-LG, at a very low relative abundance, have been found in the milk of lactating mothers who drank bovine milk. Not in any control samples. A control was positive in ELISA. This inconsistency result demonstrates that immunological methods suffer from bias.
Matangkasombut, 2017 [42]	Cow’s-milk-free diet (7 d) or not (240 mL) and cow’s-milk-free diet	*n* = 15, (non-allergic children), 9 had a history of atopic diseases, *n* = 4 (allergic children), all with a history of atopy, mature milk	0, 3, 6, 24 h and 3, 7 days	β-LG, ELISA, 0.002 ng/mL	3,80 ng/mL, 24 h, 15/15 and 4/4	Significant increases in β-LG up to 7 days after maternal consumption of cow’s milk. Breastfeeding with milk containing β-LG elicited symptoms of allergy in three of the four allergic infants. No statistic difference in the levels of β-LG in milk from atopic and non-atopic lactating mothers was found.
Zhu, 2019 [43]	Unrestricted diet	*n* = 6, healthy, mature milk	N/R	αS1-, αS2-, β-, κ-caseins, and β-LG protein/peptides. Electrophoresisprefractionation and HPLC/MSMS Data-Dependent Shotgun Analysis, N/R	N/R, 6/6	Strong evidence for the presence of intact nonhuman proteins originated mostly from bovine origin in human milk but in nM range.
Picariello, 2019 [17]	Cow’s-milk-free diet (7 d) or not (200 mL)	*n* = 1, non-atopic, mature milk	2–3 h, several days	Intact β-LG and derived peptides from β-LG and caseins, Dot-Blot (1 pg) and Western blotting, Competitive ELISA (2.1 ppm), Nanoflow-HPLC-MSMS (N/R)		Peptides from both bovine caseins and whey proteins were identified in the enriched peptide fraction of breast milk. These peptides were missing after a prolonged cow’s-milk-free diet. No intact cow’s milk gene products were detected.
Dekker, 2020 [16]	Unrestricted diet	*n* = 10, allergic, *n* = 10, nonallergic, mature milk	N/R	Peptide sequences of 29 different bovine proteins (β-LG and caseins), LC-MS/MS, N/R	N/R	A significant difference in levels of nonhuman proteinaceous molecules in human milk of allergic and nonallergic mothers has been observed. This difference can be largely attributed to sequences that match to bovine proteins β-LG and a2-HS-glycoprotein. These findings suggest that there is a difference in transfer of proteinaceous molecules through the intestinal barrier of allergic mothers, allowing dietary proteins to enter the bloodstream and ultimately the milk.

β-lactoglobulin, (β-LG); weeks (w); days (d); not reported (N/R); radioimmunoassay (RIA); gel electrophoresis in denaturing conditions (SDS-Page); Western blotting (WB); reverse phase high-performance liquid chromatography-mass spectrometry (RP-HPLC); protein purification methodologies followed microsequencing (sequencing); loading on gel, image acquisition by proteomic imaging system, in-gel tryptic digestion of proteins, and protein identification by tandem mass spectrometry (Proteomic Techniques); liquid chromatography with tandem mass spectrometry (LC/MSMS); high-performance liquid chromatography/high-resolution mass spectrometry (HPLC/HRMS); cow’s milk protein (CMP).

## Data Availability

Not applicable.

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
