# Peer review of "Cow’s Milk Antigens Content in Human Milk: A Scoping Review"

_foods, 2022, doi:10.3390/foods11121783_

Round 1
Reviewer 1 Report
The manuscript entitled „Cow’s milk antigens content in human milk: a scoping review” by Carlos Franco, Cristina Fente, Cristina Sánchez, Alexandre Lamas, Alberto Cepeda, Rosaura Leis and Patricia Regal showed the results of studies made by various authors on the presence of cow milk allergens in breast milk. This is a review of the available scientific data.
Here are some remarks that should improve.
Line 107, 108: Two authors have the same initials (C.F.). It is not clear to the reader whether this is two authors or perhaps an editorial mistake. I would like to propose that these initials be differentiated.
Line 154: αs-1casein is not a whey protein. Please, improve this sentence.
Line 155: Do the authors mean αs2-casein or αs-1casein in this place ?
Line 155: β-lg as a monomer has a molecular weight 18.6 kDa, not 36 kDa.
Line 173: From an analytical point of view, intact protein determination of intact protein by HPLC-MS/MS is not possible due to protein digestion, which is part of sample preparation.
In description 3.3. The problems with the analytical possibilities determintaion of selected proteins and/or peptides should be presented. The analysis itself causes denaturation and hydrolysis of proteins, so the question is how to indicate the presence of intact proteins (not subjected to any analytical or processing steps). Please respond to this problem.
Line 178 - 183: The authors quote Piciarelli's work two times. In the first sentence, there is an information that „ HPLC-MS/MS techniques have been used to detect intact cow’s milk proteins” and in the second one that „Picariello et al. [16] couldn't detect intact cow’s milk proteins by HPLC-MS/MS. Please make this clearer.
Line 185 i 212: The same paragraph number was used (3.4.). Consequently, subsequent paragraph numbers need to be corrected.
Line 272: According to my opinion, it is not only „The existence of the intestine-breast axis can explain presence of these non human molecules in human milk”, but the biochemical and immunological mechnisms of interactions between molecules, cells, interleukins, etc. taking place in the intestine-breast axis is very important and essential. Would you be so kind and describe it in more details?
The revised manuscript may be published in FOODS.
Author Response
Response to reviewer 1:
The authors are grateful to the referee’s comments. In the revised version of the manuscript the aspects mentioned by the reviewer have been corrected.

Reviewer 2 Report
This review mainly summarizes and discusses the clinical relevance of cow’s milk antigens content in human milk in the induction of allergic sensitization in breastfed infants. However, there is a misunderstanding in the sentence “Studying variations between mothers…and analytical methodology” in the abstract. With regard to the results, the summary of literature in 3.4, 3.5, 3.6, and 3.7 is too simplistic. Firstly, the concentration of allergens found in human milk in those literatures need to be described and summarized. Secondly, the results in these sections need to be supplemented with the literature of clinical relevance. Thirdly, the relevance of the allergen peptide level in human milk to maternal diet, lactation stage, and clinical manifestations in the infant needs to be discussed. In addition, many formatting and writing errors need to be modified.
Specific Comments:
- There are many mistakes in the reference format, for example, line 38, line 225-226, line 310-312, line 344-345, line 350, and line 353. The format of references in the text should be uniform and in accordance with the requirements of the journal of Foods.
- Line 60
“conclude” need to be modified to “concluded”.
- “cow milk” should be modified to “cow’s milk”, for example in lines 85 and line 192
- The unit of “ml” should be “mL”
- Line 138
The annotation of Table 1 needs to be left-aligned to the table.
- The single word in the table should in a line.
- The full name of “COW” should be added.
- The letter “β” with many wrongs in this text, such as lines 151-163
- Line 180
There is an extra space between “In 2019,” and “Picariello”
- Line 182
“antiß-LG” should be modified to “anti-β-LG”
- Line 198
“suggest” should be modified to “suggested”
- Line 212
“3.4” should be modified to “3.5”.
- Line 220
“3.5” should be modified to “3.6”.
- Line 242
“3.6” should be modified to “3.7”.
- Line 248
There is an extra space between “diet” and “[30]”.
- Line 263
There is an extra space between “adulthood” and “[48]”.
- Line 286 and line 293
“cow milk protein” should be “cow’s milk proteins”
- Line 323
“aS1-casein” should be “αS1-casein”.
There is an extra space between “[36,37]” and “However”.
Author Response
Response to reviewer 2
The authors are grateful to the referee’s comments. In the revised version of the manuscript the aspects mentioned by the reviewer have been corrected.

Reviewer 3 Report
A very valuable scoping review of 27 publications from 1984-2020, selected according to the PRISMA flow chart rules. The analysed publications concern the content of beta-lactoglobulin and some fragments of derived from bovine proteins in the milk of breastfeeding women before and after maternal consumption.
To determine the content of beta-LG and other bovine proteins derived in the breast milk, the researchers used the analytical methods described in subchapter 3.3.
The presence of these proteins in breast milk was analysed depending on the diet of the nursing woman and the lactation period (samples of colostrum, transient milk, and mature milk). The study group included healthy mothers, mothers with an allergic history and mothers of premature babies.
The Results and Discussion chapter provides detailed information on the content and type of native protein (beta-LG, LS1-casein) and other derivatives that were confirmed in human milk samples from 26 out of 27 examined breastfeeding mothers.
The research results confirm the transfer of cow's milk proteins after material digestion to mammary glands and then into human milk. The obtained results are especially important as they confirm that proteins, which are a potential source of allergens for a breastfed infants may pass into breast milk.
This potentially allergenic properties of bovine molecules was greater in the milk of atopic mothers and mothers of premature babies, in comparison to the milk of healthy mothers.
If early allergic sensitisation overcome early oral tolerance in breastfed infants, a temporary elimination of cow's milk proteins from the diet should be considered in mothers of these infants.
Author Response
Response to reviewer 3:
The authors are grateful to the referee’s comments.
Reviewer 4 Report
The topic of the present review is very interesting, given the importance of improving our knowledge on cow’s milk antigen content which may in turn affect tolerance mechanisms.
The review is well constructed, easy to read and provides a quite exhaustive overview of this particular field of interest and research.
My comments, as following:
Introduction: Recent reviews proposed that antigen characteristics may play a role in oral tolerance induction.
The authors rightly state that the potential induction of early oral tolerance to overcome early allergic sensitization through exposure to allergens in breast milk also
remains controversial and not yet well established. They mention the amount of food, the period that elapses between its intake and sample collection, lactation stage and lastly the allergic status of the mothers as factors potentially influencing the presence and the amount of cow’s allergens content in human milk.
Antigen characteristics however include not only dose and kinetics but also mode of food antigen consuming (e.g cooking, matrix etc) that influence the allergenicity which in turn promote or limit the accessibility of peptides to the immune system, also in breast milk.
The interactions between milk proteins and some components of the food matrix during heating seem to play the most important role in the reduction of allergenicity, limiting the accessibility of peptides to the immune system (Bavaro et al. 2019).
This issue should be included in the paragraph 3.4 and briefly discussed in the Discussion.
Paragraph 3.3
Besides to the mentioned techiniques, other existing new techniques, such as surface plasmon resonance (SPR) should be taken into account
In fact simple and label free SPR sensors have been just developed to detect casein and b-lactoglobulin (see and cite the recent review by Benedè S et al. Critical Reviews in Food Science and Nutrition, 2021)
In the Discussion section the authors focused on proteic component of human milk however, it is likely the lipids, which represent an important component of human milk, may play an important role.
It is known that large number of food allergens bind and transport lipids whose ability to modulate the immune response has been reported (Tordesillas et al. 2017; L_opez-Fandi~no 2020). I suggest to include this in the Discussion
Breast milk is a complex system; thus, it is likely that many other factors, other than proteins, present in breast milk play a role probably may explain some of observed differences among the studies.
-To further improve the review a knowledge gaps Table should be included
Author Response
Response to reviewer 4:
The authors are grateful to the referee’s comments. In the revised version of the manuscript some aspects mentioned by the reviewer have been corrected.
